# Efficacy of Immune Checkpoint Inhibitor (ICI) Rechallenge in Advanced Melanoma Patients’ Responders to a First Course of ICI: A Multicenter National Retrospective Study of the French Group of Skin Cancers (Groupe de Cancérologie Cutanée, GCC)

**DOI:** 10.3390/cancers15143564

**Published:** 2023-07-10

**Authors:** Charlée Nardin, Aymeric Hennemann, Kadiatou Diallo, Elisa Funck-Brentano, Eve Puzenat, Valentine Heidelberger, Géraldine Jeudy, Mahtab Samimi, Candice Lesage, Lise Boussemart, Lucie Peuvrel, Jacques Rouanet, Florence Brunet-Possenti, Emilie Gerard, Alice Seris, Thomas Jouary, Mélanie Saint-Jean, Marc Puyraveau, Philippe Saiag, François Aubin

**Affiliations:** 1Service de Dermatologie, Centre Hospitalier Universitaire, 25000 Besancon, France; ahennemann@chu-besancon.fr (A.H.); faubin@chu-besancon.fr (F.A.); 2Université Franche Comté, Inserm 1098 RIGHT, 25020 Besancon, France; 3Centre de Méthodologie Clinique, Centre Hospitalier Universitaire, 25030 Besancon, France; kdiallo@chu-besancon.fr (K.D.);; 4Université Paris-Saclay, UVSQ, EA4340-BECCOH, Assistance Publique–Hôpitaux de Paris (AP-HP), Hôpital Ambroise-Paré, Service de Dermatologie Générale et Oncologique, 92104 Boulogne-Billancourt, France; elisa.funck-brentano@aphp.fr (E.F.-B.); philippe.saiag@uvsq.fr (P.S.); 5Service de Dermatologie, Hôpital Robert Ballanger, 93420 Villepinte, France; valentine.heidelberger@ght-gpne.fr; 6Service de Dermatologie, Centre Hospitalier Universitaire, Hôpital Le Bocage, 21079 Dijon, France; geraldine.jeudy@chu-dijon.fr; 7Service de Dermatologie, Centre Hospitalier Universitaire, BIP 1282, INRA-Université de Tours, 37020 Tours, France; 8Service de Dermatologie, Centre Hospitalier Universitaire, 34295 Montpellier, France; candice.lesage@icm.unicancer.fr; 9Service de Dermatologie, Centre Hospitalier Universitaire, Université de Nantes, INSERM, Immunology and New Concepts in Immunotherapy, INCIT, UMR 1302, 44000 Nantes, France; 10Institut de Cancérologie de l’Ouest, 44800 Saint-Herblain, Francemelanie.saintjean@ico.unicancer.fr (M.S.-J.); 11Service de Dermatologie, Centre Hospitalier Universitaire, 63003 Clermont-Ferrand, France; jrouanet@chu-clermontferrand.fr; 12Service de Dermatologie, Hôpital Bichat AP-HP, Université Paris Cité, 75018 Paris, France; florence.brunet-possenti@aphp.fr; 13Service de Dermatologie, Centre Hospitalier Universitaire, 33075 Bordeaux, France; emilie.gerard@chu-bordeaux.fr; 14Oncologie Médicale, Centre Hospitalier, 64046 Pau, France

**Keywords:** rechallenge, retreatment, re-induction, immunotherapy, immune checkpoint inhibitor, anti-PD1, anti-CTLA-4, melanoma, response, disease control, safety

## Abstract

**Simple Summary:**

The long-term effectiveness of immune checkpoint inhibitor (ICI) rechallenge for progressive or recurrent advanced melanoma following previous disease control induced by ICI has not been well-described in the literature. The objective of our retrospective multicenter study was to evaluate the efficacy and safety of ICI rechallenge in patients with advanced melanoma who had achieved disease control with ICI in a real-life setting. Our study, which included 85 patients, predominantly rechallenged with anti-PD1 antibodies, confirms the efficacy of ICI rechallenge, with a best overall response rate of 54% and a disease control rate of 75%. Twenty-eight adverse events (AEs) were reported in 23 patients (27%), including 18 grade 1–2 AEs and 10 grade 3–4 AEs. Therefore, ICI rechallenge should be considered as a compelling therapeutic option.

**Abstract:**

Background: The long-term effectiveness of immune checkpoint inhibitor (ICI) rechallenge for progressive or recurrent advanced melanoma following previous disease control induced by ICI has not been thoroughly described in the literature. Patients and methods: In this retrospective multicenter national real-life study, we enrolled patients who had been rechallenged with an ICI after achieving disease control with a first course of ICI, which was subsequently interrupted. The primary objective was to evaluate tumor response, while the secondary objectives included assessing the safety profile, identifying factors associated with tumor response, and evaluating survival outcomes. Results: A total of 85 patients from 12 centers were included in the study. These patients had advanced (unresectable stage III or stage IV) melanoma that had been previously treated and controlled with a first course of ICI before undergoing rechallenge with ICI. The rechallenge treatments consisted of pembrolizumab (*n* = 44, 52%), nivolumab (*n* = 35, 41%), ipilimumab (*n* = 2, 2%), or ipilimumab plus nivolumab (*n* = 4, 5%). The best overall response rate was 54%. The best response was a complete response in 30 patients (35%), a partial response in 16 patients (19%), stable disease in 18 patients (21%) and progressive disease in 21 patients (25%). Twenty-eight adverse events (AEs) were reported in 23 patients (27%), including 18 grade 1–2 AEs in 14 patients (16%) and 10 grade 3–4 AEs in nine patients (11%). The median progression-free survival (PFS) was 21 months, and the median overall survival (OS) was not reached at the time of analysis. Patients who received another systemic treatment (chemotherapy, targeted therapy or clinical trial) between the two courses of ICI had a lower response to rechallenge (*p* = 0.035) and shorter PFS (*p* = 0.016). Conclusion: Rechallenging advanced melanoma patients with ICI after previous disease control induced by these inhibitors resulted in high response rates (54%) and disease control (75%). Therefore, ICI rechallenge should be considered as a relevant therapeutic option.

## 1. Introduction

Immune checkpoint inhibitors (ICI) have improved advanced melanoma patients’ treatment [1], particularly the anti-programmed death-1 (PD-1) monoclonal antibodies (mAb) nivolumab and pembrolizumab, which induced, in the first-line setting, longer progression-free survival (PFS) and overall survival (OS) than chemotherapy or ipilimumab (anti-CTLA-4 mAb) [2,3]. Depending on the treatment line [4] and length of follow-up, anti-PD-1 monotherapy was associated with an overall response rate (ORR) of 27–52% and a median PFS of 3.1–6.9 months [2,3,4,5,6] in patients without active brain metastasis. Nivolumab plus ipilimumab induces the longest PFS and OS but is associated with frequent severe adverse events (AEs) [2,6]. This combination also provides high ORR and encouraging PFS and OS data in melanoma patients with 1–4 asymptomatic intracranial metastases [7,8].

Besides toxicity or the patient’s wish, ICI therapies can also be withdrawn, with some data suggesting at least 6 months of anti-PD1 treatment when a complete response (CR) persists at radiological evaluation, or in case of prolonged partial response (PR) or stable disease (SD) after 2 years of anti-PD1 treatment [3]. However, after treatment withdrawal, melanoma recurrence has been observed in 10–15% of patients who achieved a complete response (CR) [4,5], with progressive disease (PD) being more common (32 to 50%) in patients who achieved a partial response (PR) or stable disease (SD) as their best response [4,6,7,8,9]. Such patients may undergo a second course of ICI called “rechallenge”.

There are many data regarding the tolerance of ICI rechallenge after the occurrence of treatment AEs but data are scarce regarding the efficacy of ICI rechallenge for recurrent or progressive disease after initial disease control with a previous course of ICI [10,11,12,13]. Indeed, data, including clinical trials and real-life cohorts, came from small patient groups data and the efficacy of ICI rechallenge varies [6,9,14,15,16,17,18,19,20].

Therefore, we performed a national multicenter retrospective study (REIMMUNO) to evaluate the efficacy and tolerance of ICI rechallenge in melanoma patients with progressive or recurrent advanced melanoma initially controlled with a first course of ICI.

## 2. Materials and Methods

### 2.1. Data Collection

We conducted a retrospective multicenter study in 12 French skin cancer referral departments to evaluate the efficacy and tolerance of ICI rechallenge in advanced melanoma patients. The study included all patients with confirmed advanced melanoma who received ICI treatment, achieved disease control resulting in treatment withdrawal, and subsequently experienced disease relapse (recurrence or progression) and were treated with ICI rechallenge. Data collection was completed by the co-investigators using anonymized case report forms, which were then imported into a Microsoft Excel file for analysis by the methodological and statistical department. The database was locked in July 2022, and all recorded American Joint Committee on Cancer (AJCC) stages were converted to the 8th edition for consistency.

### 2.2. Patients

Included patients had advanced melanoma (unresectable AJCC 8th stage III or stage IV disease) and a controlled disease (CR, PR or SD) obtained after a first course of ICI therapy (ipilimumab or anti-PD1 antibodies or anti-PD1 antibodies plus ipilimumab), regardless of their prior treatments, and had to be rechallenged with ICI (without any time limit between ICI withdrawal and rechallenge). Patients had to be rechallenged with ICI for recurrent or progressive disease. The terms “rechallenge” and “retreatment” were recently well differentiated [21]. Retreatment is defined as “repeated treatment with the same therapeutic class following relapse after adjuvant treatment has ended”, whereas rechallenge is “repeated treatment with the same therapeutic class following disease progression in patients who had clinical benefit with prior treatment for unresectable or metastatic disease”. We distinguished three situations: the rechallenge of the same ICI (anti-PD1 after anti-PD1 or anti-CTLA-4 after anti-CTLA-4 or the combination ICI after the combination ICI), the switch when using anti-PD1 after using anti-CTLA-4 or vice versa, and the escalation when using the combination ICI (anti-PD1 plus anti-CTLA-4) after anti-PD1 or anti-CTLA-4 monotherapy.

According to French law, this study abided by standard medical practices and did not require written informed consent nor formal approval by a national ethics committee. However, consent was obtained orally from all living patients, and the protocol was accepted by the research ethics committee. The study was conducted in accordance with the principles of the Declaration of Helsinki and the ethics committee.

Clinical characteristics of patients and melanoma at the initiation of the first ICI treatment and of rechallenges (including second rechallenge if applicable) were collected.

### 2.3. Objectives

The primary endpoint was the best overall response rate (BORR) evaluated in each center during ICI rechallenge.

Secondary endpoints were: disease control rate (DCR), duration of response (DoR), factors associated with response and disease control, tolerability of ICI rechallenge (frequency of AEs, type and severity according to the Common Terminology Criteria for Adverse Events V5), and outcomes of patients following ICI rechallenge with PFS (time from the first dose of rechallenge to PD, or death) and OS (time from ICI rechallenge to death).

Tumor response was assessed using the RECIST criteria (Response Evaluation Criteria in Solid Tumor). The best overall response is the best response recorded from the start of the treatment until disease progression/recurrence. BORR is the presence of at least one confirmed CR or confirmed PR. DCR is the presence of at least one confirmed CR or PR or SD. DoR is the length of time during which a tumor continues to respond to treatment without the cancer growing or spreading.

### 2.4. Statistical Analyses

Follow-up time was measured from the times of ICI rechallenge to last follow-up. Characteristics are presented for all patients using descriptive statistics. Categorical variables are reported as numbers and percentages, and quantitative variables as means and confidence intervals or medians and ranges. Bivarious analyses were made with Student’s T-test or Wilcoxon test according to the variable distribution. Chi-square test or exact Fisher test were performed for qualitative variable. Univariate and multivariate (when possible) logistic regressions were also performed to model treatment response. OS and PFS were analyzed with Kaplan–Meier method and Cox models. Odds ratios (OR) and hazard ratios (HR) and 95% confidence intervals (95% CIs) are presented. Variables with a *p*-value ≤ 0.2 in the univarious model were included in the multivarious model (Cox and logistic). Statistical significance was defined as *p* ≤ 0.05, and all tests were two-sided. All analyses were performed with the R statistical software v.4.1.0.

## 3. Results

### 3.1. Characteristics of Patients

Overall, 85 patients from 12 French centers rechallenged with an ICI between July 2014 and June 2021 were included. The initial characteristics of the patients and their disease at the time of rechallenge are summarized in Table 1.

The median time between ICI withdrawal and relapse of the disease was 6 months (range: 1–15). The median time between the first ICI withdrawal and rechallenge was 13 months (range: 2–60). The median duration of rechallenge was 9 months (range: 1–72).

The first ICIs used (Appendix A) were mainly anti-PD1 antibodies (80%), followed by ipilimumab (12%) or the combination treatment of ipilimumab plus nivolumab (8%). Patients were rechallenged with anti-PD1 monotherapy (93%), ipilimumab (2%) and ipilimumab plus nivolumab (5%).

### 3.2. Response to ICI Rechallenge

The best response to ICI rechallenge was CR in 30 patients (35%), PR in 16 patients (19%) and SD in 18 patients (21%), corresponding to a BORR of 54% [95%CI, 36.4–75.2] and a DCR of 75% [95%CI, 64.7–84.0]. Twenty-one patients (25%) did not respond to ICI rechallenge. The median time to best response (CR plus PR) after ICI rechallenge was 3 months (range: 1–37) (Figure 1 and Appendix A).

Among the 68 patients treated with anti-PD1 monotherapy during the first course of ICI, 64 patients were rechallenged with anti-PD1 monotherapy. CR, PR, SD and PD were observed in 25 (39%), 12 (19%), 11 (17%) and 16 (25%) patients, respectively. Four patients were rechallenged with anti-PD-1 plus anti-CTLA-4 which elicited one CR, one PR, one SD and one PD.

When we considered the three situations of ICI rechallenge, there were four escalations leading to a BORR of 50% and a DCR of 75%, eight switches (from anti-CTLA-4 to anti-PD1) leading to a BORR of 62.5% and a DCR of 87.5%, and 73 rechallenges with the same ICI class (mostly anti-PD1 to anti-PD1) leading to a BORR of 55% and a DCR of 71%.

At ICI rechallenge, 22 (26%) patients had brain metastases (BM), including 12 (55%) patients who had previously had BM and 10 (45%) who developed BM after the interruption of the first course of ICI. Among the 16 patients who had brain metastases before receiving the first ICI, 12 patients underwent stereotactic radiosurgery, with one patient also undergoing surgery. Among the 22 patients with brain metastases prior to ICI rechallenge, nine patients had received stereotactic radiosurgery. BORR to ICI rechallenge in these 22 patients with BM was 45% (six CR, four PR, five SD and seven PD).

### 3.3. Progression-Free Survival and Overall Survival

Median follow-up was 11 months (range: 1–72), and disease control obtained with ICI rechallenge initiation was maintained in 43 patients (67%) (Figure 1) with a median DoR of 5.82 months [95% CI, 2.66–9.24].

The second ICI (ICI rechallenge) was discontinued in 49 patients (58%), because of progression (*n* = 30, 35%), disease control (*n* = 11, 13%), toxicities (*n* = 6, 7%) or the patient’s wish (*n* = 2, 2%).

Median PFS after rechallenging was 21 months and median OS was not reached (Figure 2A,B). One-year and 2-year PFS were respectively 58% and 47%, and 1-year and 2-year OS were respectively 78% and 71%.

At last follow-up, 42 (49%) patients demonstrated a recurrence or progressive disease, and 23 (27%) had died.

As expected, responses to ICI rechallenge were associated with better PFS and OS (*p* < 0.0001 for both) as shown in Figure 2C,D.

### 3.4. Factors Associated with Response and Outcomes to ICI Rechallenge

In multivariate analysis, the use of a systemic treatment between the two ICI courses was the only factor independently associated with a lower response to ICI rechallenge (OR, 0.249; 95%CI, 0.061–0.860; *p* = 0.035) (Table 2).

Factors associated with PFS after rechallenging are presented in Appendix A. The factors significantly associated with a better PFS (Appendix A) in multivariate analysis were: less duration of first ICI (*p* = 0.031), no other systemic treatments between the courses (*p* = 0.016), and ≤3 metastatic sites (*p* = 0.007).

Factors associated with OS after rechallenging are presented in Appendix A. The factors significantly related to a better OS (Appendix A) in multivariate analysis were: ≤3 metastatic sites (*p* = 0.020), no corticosteroids use (*p* = 0.007), and the occurrence of toxicity (*p* = 0.025).

### 3.5. Second Rechallenge with Immune Checkpoint Inhibitors

At last follow-up, 42 (49%) patients had progressed including 21 (25%) non-responders to ICI rechallenge and 21 (25%) patients with acquired resistance to ICI rechallenge (Figure 1 and Appendix A). The median time between ICI rechallenge and progression was 5 months (range: 2–9). Among these 42 patients who had progressed after ICI rechallenge, 15 (36%) had a second ICI rechallenge with anti-PD1 alone (*n* = 7, 47%), ipilimumab plus nivolumab (*n* = 5, 33%), anti-PD1 plus anti-LAG3 (*n* = 1, 7%) or ipilimumab alone (*n* = 2, 13%). The second rechallenge led to one CR (7%), three PR (20%) and 11 PD (73%).

### 3.6. Safety of ICI Rechallenge

Tolerance to the first and second courses of ICI is shown in Appendix A. Following ICI rechallenge, we observed 28 AEs, which occurred in 23 (27%) patients after a median time of 3 months (range: 1–37). Fourteen patients (18%) developed 18 mild to moderate AEs (grade I–II). Nine patients (11%) had severe toxicities [8 grade III AEs (9%) (including two cutaneous, two liver, one gastrointestinal, one renal, one respiratory and one haematological AEs) and one gastrointestinal and endocrine grade IV AE (1%)]. Five patients (6%) developed AEs for which they interrupted the treatment.

Among the 43 patients (51%) who had AEs during the first course of ICI, 18 patients (21%) underwent AEs during ICI rechallenge. Six patients had the same recurrent AE during ICI rechallenge. Among the patients who underwent severe AEs (*n* = 19) during the first ICI, nine patients (47%) had AEs during the ICI rechallenge [including six mild/moderate AEs and three severe AEs (two cutaneous grade IV and one liver grade IV)].

## 4. Discussion

In this French retrospective multicenter study, we investigated melanoma patients who had undergone ICI withdrawal due to treatment success but subsequently experienced relapse. The rechallenge with ICI, specifically anti-PD1 (used in 93% of cases), was found to be associated with a high DCR of 75%.

The time intervals between ICI interruption and relapse, as well as between the two courses of ICI, were 6 months and 13 months, respectively. These durations were consistent with previously reported data [10,16,17,18,20,22].

In this study, we present the largest cohort of melanoma patients who were rechallenged with anti-PD1 antibodies for progressive disease after achieving disease control with ICI. Indeed, most of the available data on the efficacy of ICI rechallenge come from small patient groups, including clinical trials and real-life cohorts with a BORR ranging from 12% to 54%, and a DCR ranging from 46% to 87.5% (Table 3) [9,15,16,17,18,19,20,22,23]. The efficacy of ICI rechallenge in our cohort was high, similar to the retrospective studies [16,17,18,19,20,23] and very close to the Keynote-006 trial [15]. We observed that the BORR to ICI rechallenge reported by Betof et al. was much lower compared to other studies (12% with anti-PD1 and 33% with ipilimumab plus nivolumab) [22]. One could hypothesize that this difference could be attributed to the fact that patients had prior exposure to ipilimumab, as 45% of the patients included in the initial study population had previously received ipilimumab. The observed response rates may have been influenced by a potential selection bias towards a more challenging population, consisting of patients who experienced recurrence after ipilimumab and anti-PD1 treatment.

Regarding factors associated with the efficacy of ICI rechallenge in our study, the only independent factor associated with a poor response was the introduction of systemic treatment (including chemotherapy, clinical trials and targeted therapies) between the two courses of ICI. This factor was linked to lower PFS and tended to be associated with lower OS in the univariate analysis. Asher et al. also observed a similar result with prior treatment before rechallenge (OR = 2.8, *p* = 0.027) [17]. On the contrary, we did not observe any impact from local treatments (surgery and radiotherapy) on treatment response and patient outcomes in our study. It is known that the addition of radiotherapy has been associated with a favorable OS in patients with melanoma BM undergoing systemic therapy [25]. However, in our study, we were unable to evaluate the specific impact of radiotherapy for melanoma BM since we did not have information regarding the specific metastases that received radiotherapy.

Patients with ≤3 metastatic sites had a lower risk of recurrence and death, consistent with findings in solid tumors treated with ICI, including melanoma [26]. However, this factor was not predictive of response to rechallenge [20], likely because we included patients who had previously responded to ICI.

Furthermore, a treatment with corticosteroids before the initiation of ICI rechallenge was associated with a poorer OS. Asher et al. also reported a tendency of a lower PFS in patients treated with corticosteroids [17]. Indeed, when corticosteroids are used for supportive care or BM, the prognosis of patients under ICI is poor (HR, 2.5; 95%CI, 1.41–4.43; *p* < 0.01 and HR, 1.51; 95%CI, 1.22–1.87; *p* < 0.01 respectively) [27]. In our cohort, we did not collect information on the indication for corticosteroids but we noticed that six out of nine patients had BM suggesting that they were symptomatic.

Another factor which may impact the efficacy of ICI rechallenge is the type of ICI used. We did not find any difference between rechallenge with the same molecule, escalation or switching, possibly due to the low number of patients treated with a switch and escalation of ICI (as most patients received anti-PD1 monotherapy). Comparing the efficacy of different ICI used for rechallenge is challenging in the literature, as meta-analyses include all patients rechallenged with ICI, regardless of the reason (toxicities, progression under ICI or interruption of ICI) [28,29].

We did not find a correlation between response to the initial course of ICI (CR plus PR as compared to SD) and response to rechallenge as reported by Betof Warner et al. [22]. However, CR to rechallenge was more frequent in patients who had achieved a CR with the first course of ICI (45%, 28% and 10% if CR, PR and SD with the 1st course of ICI respectively).

As expected, the types of responses (CR, PR, SD) to ICI rechallenge were associated with increased survival (PFS and OS) after rechallenge, consistent with findings from clinical trials of patients treated with ICI [14,15] and with ICI rechallenge [17].

Furthermore, as previously observed in patients receiving the first course of ICI, we observed that rechallenged patients who had experienced toxicities presented increased OS compared to patients without toxicities [24,30,31,32].

Finally, we found that BOR to the second ICI rechallenge (27%) was much lower than the rate observed after the first ICI rechallenge (50%) and the first course of ICI (88%). The efficacy of ICI rechallenge seems to decrease with repeated rechallenge treatments, as previously reported in other studies, including meta-analyses [16,18,22,28,29].

Regarding the safety of ICI rechallenge, the most frequent AEs and the frequency of severe AEs (11%) were consistent with the literature [6,7,8,9,10,33,34]. AEs were less frequent during the rechallenge compared to the first course of ICI (33% versus 67%) as previously reported [10,35,36]. Among patients who had experienced severe AEs (*n* = 19) with the first ICI, only three patients (16%) experienced severe AEs during the rechallenge. This may be due to the early management of toxicities upon resumption of ICI. Thus, ICI rechallenge demonstrated a good safety profile and patients who experienced AEs during the first ICI (including severe AE), are not likely to present the same severe AE.

The strength of this study is that we report the largest cohort of melanoma patients rechallenged with anti-PD1 antibodies after achieving disease control with ICI. Additionally, we provide data on a subsequent (second) rechallenge for the first time.

Our study has some limitations. Firstly, the retrospective nature of the study could introduce memory bias and missing data. Secondly, our data only came from voluntary participating centers, as there is no available national exhaustive data. Factors associated with treatment response, PFS, and OS were not consistently the same, suggesting that there may be different predictive and prognostic factors for ICI rechallenge. These differences may also be due to a lack of statistical power.

## 5. Conclusions

Our study demonstrates that ICI rechallenge for disease progression or recurrence after achieving disease control with a previous course of ICI resulted in a high rate of disease control (75%) and should be considered in such situations.

ICI rechallenge and the factors associated with response should be further investigated as ICI rechallenge is likely to become more frequent in the future, such as after adjuvant or neoadjuvant treatments.

## Figures and Tables

**Figure 1 cancers-15-03564-f001:**
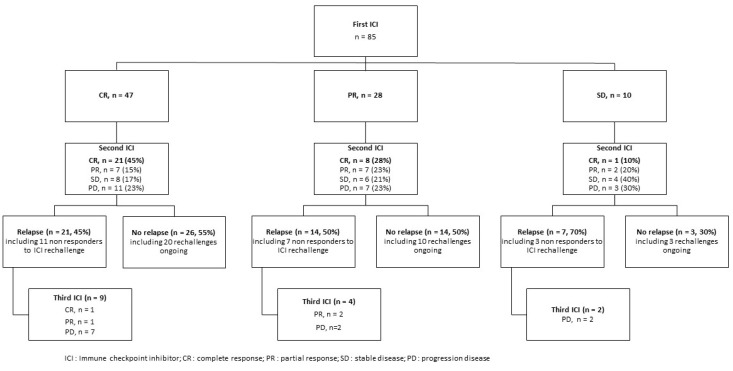
Consort flow chart according to the response over the courses of ICI (CR, PR and SD).

**Figure 2 cancers-15-03564-f002:**
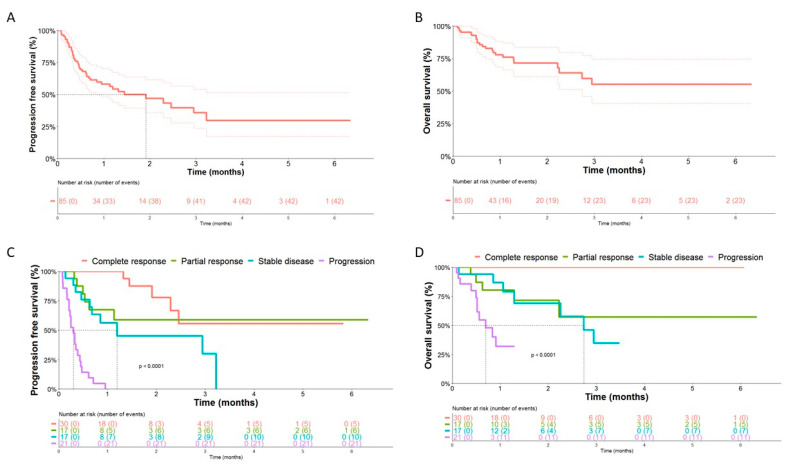
Progression-free survival (**A**) and overall survival (**B**) in the whole cohort with the associated confidence envelope (lighter dotted curves) and PFS (**C**) and OS (**D**) according to response to ICI rechallenge.

**Table 1 cancers-15-03564-t001:** Characteristics of melanoma patients at the initiation of the second course of ICI (ICI rechallenge).

	Total, *n* = 85, 100%
**Age (years)**	
Median (range)	72 (30–89)
**Gender**	
Male (%)	47 (55)
Female (%)	38 (45)
**ECOG performance status**	
0/1 (%)	81 (95)
≥2 (%)	4 (5)
**Autoimmune disease**	
No (%)	79 (93)
Yes (%)	6 (7)
**Type of primary melanoma**	
Cutaneous (%)	70 (82)
Mucosal (%)	6 (7)
Uveal (%)	3 (4)
Unknown (%)	6 (7)
**Breslow index (mm)**	
Median range	3.3 (0.5–25)
Unknown	17 (20)
**Ulceration**	
No (%)	33 (39)
Yes (%)	28 (33)
Unknown (%)	24 (28)
**Mutation**	
BRAF V600 E/K (%)	19 (22)
BRAF no V600 (%)	1 (1)
NRAS (%)	27 (32)
CKIT (%)	3 (4)
GNAQ (%)	1 (1)
Unknown (%)	34 (40)
**AJCC stage**	
IIIC (%)	7 (8)
IIID (%)	3 (4)
IV (%)	75 (88)
**Number of metastatic sites**	
≤3 (%)	48 (57)
>3 (%)	37 (43)
**Localization of metastasis**	
Cutaneous (%)	34 (40)
Lymph node (%)	33 (39)
Lung (%)	17 (20)
Liver (%)	11 (13)
Brain (%)	22 (26)
Other (%)	14 (16)
**Elevated LDH level**	
No (%)	55 (65)
Yes (%)	11 (13)
Unknown (%)	19 (22)
**Treatment between the two ICI courses**	
Surgery (%)	8 (9)
Radiotherapy (%)	10 (12)
Targeted therapy (%)	6 (7)
Chemotherapy (%)	6 (7)
Protocol (%)	5 (6)
**Second course of ICI (ICI rechallenge)**	
Rechallenge (%)	73 (86)
Escalation (%)	4 (5)
Switch (%)	8 (9)

AJCC: American Joint Committee on Cancer; ECOG: Eastern Cooperative Oncology Group; ICI: immune checkpoint inhibitor; LDH: lactate dehydrogenase.

**Table 2 cancers-15-03564-t002:** Univariate and multivariate logistic regressions for factors associated with the response to ICI rechallenge.

	Univarious Model			Multivarious Model		
Variable	OR	95% CI	*p*-Value	OR	95% CI	*p*-Value
**Age (years)**	0.99	0.95, 1.022	0.462			
**Gender**			0.665			
Male	1	—				
Female	1.210	0.512, 2.889				
**ECOG performance status (ICI rechallenge)**			0.828			
0/1	1	—				
≥2	0.800	0.092, 6.933				
**Type of primitive tumor**			0.461			
Other	1	—				
Cutaneous	1.524	0.494, 4.797				
**BRAFV600 mutation**			0.436			
No	1	—				
Yes	1.518	0.541, 4.533				
**Response to 1st ICI**			0.100			
Complete response + Partial response	1	—				
Stable disease	0.302	0.061, 1.178				
**Duration of 1st ICI (months)**	0.98	0.946, 1.022	0.403			
**Time to relapse after 1st ICI (months)**			0.725			
≤6 months	1	—				
>6 months	1.188	0.451, 3.120				
**Systemic treatment between the ICI courses**			0.020			0.035
No	1	—		1	—	
Yes	0.228	0.058, 0.742		0.249	0.061, 0.860	
**Local treatment between the ICI courses**			0.980			
No	1	—				
Yes	1.014	0.356, 2.960				
**Number of metastatic sites (ICI rechallenge)**			0.521			
**˃3**	1	—				
**≤3**	1.326	0.559, 3.165				
**Brain metastasis (ICI rechallenge)**			0.283			
No	1	—				
Yes	0.586	0.216, 1.554				
**Elevated LDH level (ICI rechallenge)**			0.036			0.061
No	1	—		1	—	
Yes	0.440	0.113, 1.641		0.392	0.098, 1.505	
Unknown	0.244	0.075, 0.718		0.278	0.083, 0.858	
**Corticosteroids (ICI rechallenge)**			0.174			
No	1	—				
Yes	0.364	0.072, 1.486				
**Second course of ICI (ICI rechallenge)** Rechallenge Escalation Switch	1 0.825 1.375	— 0.095, 7.179 0.314, 7.104	0.896			
**Toxicity (ICI rechallenge)**			0.890			
No	1	—				
Yes	1.071	0.409, 2.859				

CI: Confidence interval; ECOG: Eastern Cooperative Oncology Group; ICI: immune checkpoint inhibitor; LDH: lactate dehydrogenase; OR: odds ratio. Only factors which are significant are shown in multivariate analysis.

**Table 3 cancers-15-03564-t003:** Tumor responses and survival to ICI rechallenge (second course of ICI) in melanoma patients in the literature.

Study	Number of Patients	1st ICI Antibodies	Response to 1st ICI	ICI Antibodies for Rechallenge	Response to ICI Rechallenge	Median Survival (Months)
**Keynote-006** [15]	13	PB	BORR 92% DCR 100%	PB	BORR 54% DCR 77%	NA
**Keynote-001** [9]	4	PB	BORR 100%	PB	BORR 25% DCR 50%	NA
**Betof Warner et al. (2019)** [22]	41	Anti-PD1 *	BORR 68% DCR 100%	Anti-PD1 63% IPI + NIVO 37%	Anti PD1: BORR 12% DCR NA IPI + NIVO: BORR 33% DCR NA	NA
**Jansen et al. (2019)** [16]	19	Anti-PD1 (PB 90%, NIVO 10%)	BORR 79% DCR 100%	Anti-PD1 (PB 79%, NIVO 21%)	BORR 32% DCR 58%	NA
**Whitman et al. (2020)** [18]	21	Anti-PD1	BORR 67% DCR 100%	Anti-PD1	BORR 48% DCR 62%	PFS: 9.9 OS: 30
**Pokorny et al. (2021)** [23]	8	Anti-PD1 (PB 92% NIVO 8%)	BORR 79% DCR 100%	Anti-PD1	BORR 50% DCR 87.5%	NA
**Asher et al. (2021)** [17]	21	Anti-PD1 81% IPI + NIVO 19%	No PD (DCR = 100%)	Anti-PD1 90% IPI + NIVO 5% IPI 5%	BORR 47% DCR 68%	NA
**Dutheil et al. (2021)** [19]	13	NA	BORR 100%	NA	BORR 38% DCR 46%	NA
**Van Zeijl et al. (2022)** [20]	27	Anti-PD1	BORR 69% DCR 100%	Anti-PD1	BORR 30% DCR 63%	NA
**Nardin et al. (2022)** [24]	85	Anti-PD1 80% (PB 48, NIVO 32%) IPI + NIVO 8% IPI 12%	BORR 88% DCR 100%	Anti-PD1 (93%) (PB 52%, NIVO 41%) IPI + NIVO 5% IPI 2%	BORR 54% DCR 75%	1-yOS: 78% PFS: 21

BORR: best overall response rate; DCR: disease control rate; ICI: immune checkpoint inhibitor; IPI: Ipilimumab; NA: not available; NIVO: Nivolumab; ORR: overall response rate; OS: overall survival; PD: progressive disease; PFS: progression-free survival; PB: Pembrolizumab; 1-yOS: one-year overall survival. * Patients treated with anti-PD1 may have had prior IPI.

## Data Availability

The data can be shared up on request.

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
