# Peer review of "Efficacy of Immune Checkpoint Inhibitor (ICI) Rechallenge in Advanced Melanoma Patients’ Responders to a First Course of ICI: A Multicenter National Retrospective Study of the French Group of Skin Cancers (Groupe de Cancérologie Cutanée, GCC)"

_cancers, 2023, doi:10.3390/cancers15143564_

Round 1

Reviewer 1 Report

Dear authors,

first of all I want to congratulate the authors to this interesting work. The manuscript is interesting and in the scope of the journal but there are some changes required.

In general, the language is good - except from the discussion part - but in all parts of the abstract and manuscript, there are minor spelling mistakes or too many or too few blank spaces - please revise 

Abstract: Twenty-eight adverse events (AEs) were observed in 23 patients 49 (27%), including 13 grade 1-2 (15%) and 9 grade 3-4 (10%) AEs - how did you calculate the percentages?

Results: could you provide Fig 1 in better resolution (pdf)?

Regarding the patients with BM, could you insert a part about the treatment of the BM if they were treated with additional (local) therapy and their results?

Tbl 1: there is a formatting error before AJCC

Fig 2: it is hard to read this Fig, because it is not vertical and also in this Figure the resolution needs to be better. What are the lighter (dotted) curves in A and B referring to?

Tbl 2: you have considered systemic treatment in between the cycles as a factor - please do consider also local treatment / radiotherapy as a factor.

23 patients (27%) underwent 28 AEs - is somehow misleading, suggest "23 patients underwent AEs, in total we observed 28 AEs among those" or else.

Five AEs (6%) - the 6% are referring to what?

Discussion

This section needs English language editing.

Including also local treatment / radiotherapy as a factor for regression analysis, please insert a short paragraph on this topic. You may cite: Franklin,C doi: 10.1136/jitc-2022-004509; Trommer,M doi: 10.3390/cancers14051240; Interno,V doi: 10.3390/cancers15051542. 

"The time-periods between ICI interruption and relapse and between the 2 courses of ICI were 6 months and 13 months respectively, which were close to that reported data" - I don't really get the meaning of this sentence.

"patients’ groups" - I suggest "patient groups"

Tbl 3: I suggest transferring table 3 and the whole paragraph about the current literature to results as a literature review you have done before executing your study. 

In general, the language is good - except from the discussion part, this needs some language editing.

Author Response

Dear reviewer,

Please find below our responses to your comments.

Reviewer 1

Dear authors,

first of all I want to congratulate the authors to this interesting work. The manuscript is interesting and in the scope of the journal but there are some changes required.

In general, the language is good - except from the discussion part - but in all parts of the abstract and manuscript, there are minor spelling mistakes or too many or too few blank spaces - please revise.

Response: Thank you for this comment. This point has been addressed in all parts of the manuscript.

Abstract: Twenty-eight adverse events (AEs) were observed in 23 patients (27%), including 13 grade 1-2 (15%) and 9 grade 3-4 (10%) AEs - how did you calculate the percentages?

Response: Thank you for this comment. There was an error. The sentence has been changed (pages 1 and 2).

The percentages of adverse events are calculated with the number of patients. "Twenty-eight adverse events (AEs) were observed in 23 patients (27%), including 18 grade I-II AEs in 14 patients (16%) and 10 grade III-IV AEs in 9 patients (11%).”

Results: could you provide Fig 1 in better resolution (pdf)?

Response : Thank you for rising this point. The figure has been changed (page 6).

Regarding the patients with BM, could you insert a part about the treatment of the BM if they were treated with additional (local) therapy and their results?

Response: Thank you for this comment. This has been added (page 7).

“Among the 16 patients who had brain metastases before receiving the first immune checkpoint inhibitor, 12 patients underwent stereotactic radiosurgery, with one patient also undergoing surgery. Among the 22 patients with brain metastases prior to immune checkpoint inhibitor rechallenge, 9 patients had received stereotactic radiosurgery”.

Unfortunately, we do not have information regarding the effectiveness of local treatment for brain metastases.

Tbl 1: there is a formatting error before AJCC

Response: Thank you for pointing out this formatting error. This has been changed (page 9).

Fig 2: it is hard to read this Fig, because it is not vertical and also in this Figure the resolution needs to be better. What are the lighter (dotted) curves in A and B referring to?

Response: Thank you for rising this point. The figure has been changed with a better resolution. The lighter curves are referring to the associated confidence envelope. This has been added page 11.

Tbl 2: you have considered systemic treatment in between the cycles as a factor - please do consider also local treatment / radiotherapy as a factor.

Response : Thank you for this comment. This has been added in the Table 2 and supplemental tables 2 and 3.

The use of a local treatment (surgery or radiation therapy) between the 2 courses of ICI was not associated witth response to ICI rechallenge [OR, 1.014 95%CI, 0.356-2.960; P = 0.980], neither associated with PFS ([HR, 1.965; 95%CI,0.997-3.872 ; P= 0.051] in univariate analysis and [HR, 1.563; 95%CI, 0.762-3.206; P = 0.223] in multivariate analysis) nor OS [HR, 0.767; 95%CI, 0.260-2.262; P = 0.631].

23 patients (27%) underwent 28 AEs - is somehow misleading, suggest "23 patients underwent AEs, in total we observed 28 AEs among those" or else.

Response: Thank you for this comment. This sentence is misleading. We have changed it “Following ICI rechallenge, we observed 28 AEs, which occurred in 23 (27%) patients after a median time of 3 months (range: 1-37) page 14.

Five AEs (6%) - the 6% are referring to what?

Response: Thank you for this comment.

In the sentence « Five AEs (6%) led to treatment interruption » refers to 5 patients.  

Indeed, 5 patients out of 85 patients (6%) had AEs for which they interrupted the treatment : one patient with a grade III gastrointestinal AE, one patient with a grade III cutaneous AE, one patient with grade II rhumatological and pulmonary AEs, one patient with grade II rhumatological and gastrointestinal Aes and one patient with grade II endocrine and haematological AEs.

This sentence has been replaced by the following sentence : « Five patients (6%) developed AEs for which they interrupted the treatment » page 14.

Discussion

This section needs English language editing.

Response: Thank you for this comment, this point has been addressed throughout the manuscript.

Including also local treatment / radiotherapy as a factor for regression analysis, please insert a short paragraph on this topic.

You may cite:

Franklin,C doi: 10.1136/jitc-2022-004509;

Trommer,M doi: 10.3390/cancers14051240;

Interno,V doi: 10.3390/cancers15051542. 

Response: Thank you for this comment. We have added a paragraph on local treatment in the discussion with the reference page 15.

Between the two courses of ICI, 18 patients had a local treatment, (8 patients underwent surgery and 10 radiation therapy).

We did not observe any impact of local treatments (surgery and radiotherapy) on treatment response and patient outcomes in our study. It is known that the addition of radiotherapy has been associated with a favorable OS in patients with melanoma BM undergoing systemic therapy (Franklin, JITC 2022). However, in our study, we were unable to evaluate the specific impact of radiotherapy for melanoma BM since we did not have information regarding the specific metastases that received radiotherapy.

"The time-periods between ICI interruption and relapse and between the 2 courses of ICI were 6 months and 13 months respectively, which were close to that reported data" - I don't really get the meaning of this sentence.

Response: Thank you for this comment. The purpose is to highlight that recurrence and rechallenge were done soon after the interruption of the treatment. The sentence has been rephrased page 14.

“The time intervals between the interruption of ICI and relapse, as well as between the two courses of ICI, were approximately 6 months and 13 months, respectively, which align with the reported data”.

"patients’ groups" - I suggest "patient groups"

Response: Thank you for this comment. We have changed it pages 3 and 14.

Tbl 3: I suggest transferring table 3 and the whole paragraph about the current literature to results as a literature review you have done before executing your study. 

Response: Response: Thank you for this comment. It is true that the review of the literature was performed before this study; however, we put the table at the end of the discussion to show in one single table the synthesis of the literature with our study so that the readers can compare our results with the existing results.

Comments on the Quality of English Language

In general, the language is good - except from the discussion part, this needs some language editing.

As mentioned above, this point has been addressed throughout the manuscript.

Reviewer 2 Report

This study reports results  of a large cohort of melanoma patients rechallenged with anti-PD1 antibodies after a previous control of the disease with ICI and report for the first time data of a subsequent (second) rechallenge. The results show that ICI rechallenge for progression/recurrence after a disease control with a previous course of ICI induced a high rate of disease control (75%) and low toxicity.

It is an extensive study, well executed and reported

Author Response

Reviewer 2

This study reports results  of a large cohort of melanoma patients rechallenged with anti-PD1 antibodies after a previous control of the disease with ICI and report for the first time data of a subsequent (second) rechallenge. The results show that ICI rechallenge for progression/recurrence after a disease control with a previous course of ICI induced a high rate of disease control (75%) and low toxicity.

It is an extensive study, well executed and reported.

Response: Thank you for your comment. We have made improvements to the English and addressed the corrections requested by the other reviewers in the revised manuscript.

Reviewer 3 Report

The primary endpoint of this retrospective study could inform and help the clinical decision in the treatment of recurrent melanoma. Despite the extensive efforts of research, at the moment, especially for BRAF wt melanomas we have (avoiding considering chemotherapy) only two drugs and, two combinations approved: anti-PD1, anti CTLA-4 and, nivolumab+ anti-CTLA-4 or plus relatlimab, with very few data on their activity and efficacy after rechallange/retreatment.

This multicenter retrospective analysis of the GCC reflects so far the largest real world experience in this field. I agree with the authors that still the numbers are not enough to provide solid values in factors associated with ORR, median PFS, and OS.  Despite that, I would invite them to add the results of the  multivariate analysis of factors significantly related to a better median PFS (supplemental table  2): duration of first ICI, no other systemic treatments between the courses, and 3 metastatic sites, as well factor significantly correlated with  a better median OS (supplemental table 3): 3 metastatic sites, no corticosteroids use and, the occurrence of toxicity

Author Response

Reviewer 3

The primary endpoint of this retrospective study could inform and help the clinical decision in the treatment of recurrent melanoma. Despite the extensive efforts of research, at the moment, especially for BRAF wt melanomas we have (avoiding considering chemotherapy) only two drugs and, two combinations approved: anti-PD1, anti CTLA-4 and, nivolumab+ anti-CTLA-4 or plus relatlimab, with very few data on their activity and efficacy after rechallange/retreatment.

This multicenter retrospective analysis of the GCC reflects so far the largest real world experience in this field. I agree with the authors that still the numbers are not enough to provide solid values in factors associated with ORR, median PFS, and OS.  Despite that, I would invite them to add the results of the  multivariate analysis of factors significantly related to a better median PFS (supplemental table  2): duration of first ICI, no other systemic treatments between the courses, and ≤ 3 metastatic sites, as well factor significantly correlated with  a better median OS (supplemental table 3): ≤ 3 metastatic sites, no corticosteroids use and, the occurrence of toxicity

Response: Thank you for rising this point.

We have added data on the factors associated with outcomes in the Results page 13:

“The factors significantly associated with a better PFS (supplemental table 2) in multivariate analysis were : less duration of first ICI (P=0.031), no other systemic treatments between the courses (P=0.016), and ≤ 3 metastatic sites (P=0.007). The factors significantly related to a better OS (supplemental table 3) in multivariate analysis were: ≤ 3 metastatic sites (P=0.020), no corticosteroids use (P=0.007) and the occurrence of toxicity. (P=0.025).”

Reviewer 4 Report

This paper described the effectiveness of rechallenge ICI use after the failure of first-line ICI with once a disease control.

Page 11, line 253-: The authors stated "In series including exclusively 254 patients who had a controlled disease with the first ICI, the efficacy of ICI rechallenge was 255 close to the efficacy found in our cohort (ORR ranging from 32 to 48 % and DCR ranging 256 from 46 to 68%) [16, 18-20]."  However, although similar reports are marked with an asterisk in Table 3, references other than references 16, 18, 19, and 20 (Pokorny et al., Asher et al.) are also marked with an asterisk. This differs from the text.

Table 3: The reference number should be added (for example: Jansen et al16.). 

Table 3: I think that the references which did not focus on the rechallenge ICIs for the patients who had a controlled disease with the first ICI should be removed, because the study design is different from the present study.  The citation of such studies makes no sense.

Reference 19: The reference format is different from the other references. Please revise it.

English quality is acceptable but plain.

Author Response

Reviewer 4 :

This paper described the effectiveness of rechallenge ICI use after the failure of first-line ICI with once a disease control.

Page 11, line 253-: The authors stated "In series including exclusively 254 patients who had a controlled disease with the first ICI, the efficacy of ICI rechallenge was 255 close to the efficacy found in our cohort (ORR ranging from 32 to 48 % and DCR ranging 256 from 46 to 68%) [16, 18-20]."  However, although similar reports are marked with an asterisk in Table 3, references other than references 16, 18, 19, and 20 (Pokorny et al., Asher et al.) are also marked with an asterisk. This differs from the text.

Response: Thank you for this comment. The sentence has been changed to include all the reports.

Of note, we extracted data from the article by Betof et al., specifically focusing on patients who were rechallenged after achieving previous disease control with anti-PD1 treatment. 

Indeed, most of the available data on the efficacy of ICI rechallenge come from small patient groups, including clinical trials and real-life cohorts with a BORR ranging from 12% to 54%, and a DCR ranging from 46% to 87.5% (Table 3) [7,15-20,22,23].

Table 3: The reference number should be added (for example: Jansen et al16.).

Response: The references have been added page 15.

Table 3: I think that the references which did not focus on the rechallenge ICIs for the patients who had a controlled disease with the first ICI should be removed, because the study design is different from the present study.  The citation of such studies makes no sense.

Response: Thank you for rising this point. We have removed them page 15 and from the references.

Reference 19: The reference format is different from the other references. Please revise it.

Response: Thank you for noticing that. This has been changed page 19.

Comments on the Quality of English Language

English quality is acceptable but plain.

Response: Thank you for your comment. This point has been addressed throughout the manuscript

Round 2

Reviewer 1 Report

Thank you very much for addressing the comments and suggestions I have raised.

Reviewer 4 Report

I have no further comments.